# Individual Evaluation of the Common Extensor Tendon and Lateral Collateral Ligament Improves the Severity Diagnostic Accuracy of Magnetic Resonance Imaging for Lateral Epicondylitis

**DOI:** 10.3390/diagnostics12081871

**Published:** 2022-08-02

**Authors:** Kazuhiro Ikeda, Takeshi Ogawa, Akira Ikumi, Yuichi Yoshii, Sho Kohyama, Reimi Ikeda, Masashi Yamazaki

**Affiliations:** 1Department of Orthopedic Surgery, Kikkoman General Hospital, Noda 278-0005, Chiba, Japan; sho_kohyama_1025@tsukuba-seikei.jp; 2Department of Orthopedic Surgery, Faculty of Medicine, University of Tsukuba, Tsukuba 305-8577, Ibaraki, Japan; ikumi@tsukuba-seikei.jp (A.I.); masashiy@tsukuba-seikei.jp (M.Y.); 3Department of Orthopedic Surgery, Mito Medical Center, Mito 311-3193, Ibaraki, Japan; ogawat@tsukuba-seikei.jp; 4Department of Orthopedic Surgery, Tokyo Medical University Ibaraki Medical Center, Ami 300-0395, Ibaraki, Japan; yyoshii@tsukuba-seikei.jp; 5Department of Orthopedic Surgery, Moriya Daiichi General Hospital, Moriya 302-0102, Ibaraki, Japan; r.ikeda0418@tsukuba-seikei.jp

**Keywords:** lateral epicondylitis, magnetic resonance imaging, severity diagnosis, diagnostic accuracy, MRI, high-resolution MRI

## Abstract

The effectiveness of magnetic resonance imaging for diagnosing lateral epicondylitis severity is controversial. We aimed to verify whether individual evaluations of the common extensor tendon and lateral collateral ligament would improve the severity diagnostic accuracy of magnetic resonance imaging for lateral epicondylitis. We obtained coronal images of the lateral elbow in three groups: healthy, clinically mild, and clinically severe. We used our scoring system for evaluation using combined and individual methods. We developed the receiver operating characteristic curve for diagnosis using the scores of the healthy and mild groups and that for severity diagnosis using the scores of the mild and severe groups. The scores, in decreasing value, were those of the severe, mild, and healthy groups, with a significant difference in both methods. The curve for diagnosis showed an area under the curve of 0.85 for the combined evaluation and 0.89 for the individual evaluation, without a significant difference between the methods (*p* = 0.23). The curve for severity diagnosis showed an area under the curve of 0.69 for combined and 0.81 for individual evaluation, with a significant difference between the methods (*p* = 0.046). Individual evaluation of the common extensor tendon and lateral collateral ligament improved the severity diagnostic accuracy of lateral epicondylitis.

## 1. Introduction

Lateral epicondylitis (LE) is tendinopathy of the common extensor tendon (CET) of the forearm [1,2], with an estimated prevalence of 1.1–4.9% in the general population [3,4,5,6]. Recent studies reported that the lateral collateral ligament (LCL) of the elbow was also injured along with the CET in LE [7,8,9]. 

Imaging studies can detect this degeneration and damage to the CET/LCL complex; ultrasonography and magnetic resonance imaging (MRI) are used in daily examinations. Ultrasonography depicts edematous changes and degeneration at the CET/LCL complex as hypointense changes or thickening, with a sensitivity rate of 64% and specificity rate of 82% [10]. However, the accuracy of ultrasonography depends on the examiner’s experience, resulting in low inter-examiner reliability [11]. The true value of ultrasonography lies in its simultaneous use in diagnosis and treatment, rather than its diagnostic ability alone. Ultrasonography is used as an essential tool for LE in accurate injection therapy and percutaneous needling [12,13,14].

Meanwhile, the validity of MRI for LE has been controversial. MRI of patients with LE shows high-signal changes in the CET/LCL complex reflecting degeneration or rupture [7,15,16]. Due to its excellent imaging quality, the inter-observer reliability is high [7,15]. However, the diagnosis of LE was based solely on physical examination findings [17,18]. MRI was only a supplemental examination for patients who had not responded to conservative treatment [1,17,19]. Although MRI is performed to diagnose pathogenesis and severity, in addition, MRI reflects asymptomatic structural abnormalities and degeneration [20,21]. Because of this low specificity, some studies have reported a negative relationship between MRI findings and clinical severity [22,23]. Conversely, more recent studies have reported positively, as MRI resolution has improved [7,15]. Despite such conflicting results, comparing the literature has proven difficult because there is no method for standard quantitative evaluation. For example, some papers evaluate the CET and the LCL together [23], while others evaluate them individually [7,15]. Furthermore, no studies have demonstrated the clinical validity for their MRI scoring. As a clinical indicator for LE, the validity of MRI remained unknown.

Based on these findings, we felt the need to demonstrate the clinical validity of MRI in the diagnosis of LE. We hypothesized that individual evaluation of the CET and the LCL would contribute to the accuracy of the assessment of severity. This study aimed to verify this hypothesis using high-resolution MRI.

## 2. Materials and Methods

### 2.1. Study Design and Participants

The study protocol conforms to the principles outlined in the 1964 Declaration of Helsinki. The study was approved by the institutional review boards of the institutions involved in this study: Mito Kyodo General Hospital (Study Number: 16–25, approved 7 September 2016) and Takahagi Kyodo General Hospital (Study Number: 10, approved 3 March 2021). We obtained written informed consent from all participating patients.

This was a case–control study. The inclusion criteria were LE patients with high-resolution MRI in our hospital and healthy adults without any history of elbow disorder. Exclusion criteria were a history of elbow trauma, elbow osteoarthritis (Kellgren–Lawrence classification 2 or higher), osteochondritis dissecans of the humeral capitellum, and rheumatoid arthritis. Our medical database identified 366 LE patients diagnosed from January 2013 to December 2020. We excluded 258 patients who did not undergo MRI, 7 patients with inappropriate MRI, and 1 patient with rheumatoid arthritis. We reviewed the electronic medical records of the remaining 100 candidates and classified them according to their clinical severity at the time of MRI; patients with a Nirschl phase rating scale of III or IV [18] were assigned to the clinically mild group (M group), and those with a score of V to VII were assigned to the clinically severe group (S group). We included 100 LE patients, of whom 41 constituted the M group and 59 the S group. Moreover, we included 30 volunteers for the healthy group (H group). The volunteers were medical coworkers in our hospital, and we selected them to match the LE patients in gender and age. The subjects comprised a total of 130 cases (median age, 49 years; age range, 23–78; 64 males, 66 females) (Figure 1).

Two upper-extremity orthopedic surgeons with 23 and 28 years of experience, respectively, made all clinical decisions for patients with LE. The diagnosis of LE was based on physical findings, positive Thomsen tests or middle finger tests, and tenderness at the lateral epicondyle [17,18,24]. Initially, we provided conservative treatment for all patients. We prescribed occupational therapy and elbow bands for all patients. Steroid injections were administered to patients, once extra-articularly and once intra-articularly, who did not respond to occupational therapy or orthotics. Some patients had received multiple steroid injections at their previous medical institutions. MRI was performed for patients who were refractory to the aforementioned conservative treatment for at least one month and had severe activity limitation due to pain, (i.e., Nirschl phase rating scale III or higher) [18].

### 2.2. MRI Protocol and Definition of the Structures

We used a clinical 3.0-Tesla imager (Magnetom Symphony, SIEMENS, Munchen, Germany) with a small-diameter surface coil (Loop Flex Coil, SIEMENS) above the lateral epicondyle of the humerus. We placed the patients’ elbows in the center of the MRI scanner, with the elbow extended beside the trunk, and the forearm supinated. We obtained a coronal section of the lateral aspect of the elbow under the following three sequences: T2*-weighted images (T2*WI) using the gradient echo to evaluate synovial folds, proton-density-weighted images (PDWI) using the high-speed spin echo to recognize the morphology of the CET/LCL complex attachment, and T2 fat-saturated weighted images (T2FSWI) to evaluate the severity of LE (Table 1).

Our protocol provided a clear and enlarged view of the lateral aspect of the elbow; we could recognize the CET and the LCL as rather isolated structures. Furthermore, we used bony landmarks to define the CET and the LCL independently (Figure 2) [25,26]. In the coronal MRI images, we defined the CET as the structure attached to the superior tubercle or the epicondylar ridge and the LCL as the structure attached to the intertubercular sulcus or the inferior surface of the posterior tubercle.

### 2.3. MRI Scoring and Evaluation

We chose T2FSWI for MRI scoring because PDWI and T2*WI had short echo times and may have overestimated the findings due to the magic-angle phenomenon (Figure 3) [27,28,29]. 

With reference to the previous literature [30,31], we created an MRI scoring scale which evaluated the strength and extent of signal changes within a coronal section on a scale of 0 to 4. The region of interest for our MRI scoring was the CET and the LCL between the articular surface of the radial head and lateral epicondyle of the humerus. Using this MRI scoring, we performed two patterns of MRI evaluation. The combined evaluation method evaluated the CET and the LCL together on a scale of 0–4. In contrast, the individual evaluation method evaluated the CET and the LCL individually on a scale of 0–4 and subsequently added the individual scores for a total score of 0–8 (Figure 4).

Two examiners independently assessed the images: an orthopedic surgeon (examiner 1) and a hand surgeon (examiner 2) with 9 and 23 years of experience, respectively. The examiners repeated the image analysis twice, with the second analysis being performed one month after the initial analysis. In the MRI evaluation, a third person blinded any clinical data and randomized the MR images.

### 2.4. Statistical Analysis

We adopted the values measured by examiner 1 for further analysis. We performed the Shapiro–Wilk test for each evaluation item as a normality test, and none of them followed a normal distribution.

We compared all of the groups’ collected variables, including clinical characteristics and MRI scores. We used the chi-squared test for categorical variables, the Mann–Whitney *U* test for continuous variables between two groups, and the Kruskal–Wallis test for comparison among three groups. When the Kruskal–Wallis test showed a significant difference, we performed Scheffe’s multiple comparison procedure. In cases of missing data for clinical characteristics, we replaced the data with the median scores of the other patients in the same group.

We created the receiver operating characteristic curve (ROC curve) for the diagnosis of LE from the MRI scoring of the H and M groups, and the ROC curve for the diagnosis of severity from the M and S groups. 

We used Fleiss’ kappa analysis to evaluate intra-observer and inter-observer reliability for the entire MRI scoring process. The interpretation of the kappa coefficient was defined as follows: 0.81–1.00 = excellent, 0.61–0.80 = good, and 0.41–0.60 = fair.

In principle, we set the level of statistical significance as *p* < 0.05. In performing the chi-square test among three groups, we corrected the significance level with Bonferroni’s method (i.e., *p* < 0.016).

We performed all statistical analyses using Bellcurve for Excel version 3.20 (SSRI Co., Tokyo, Japan).

### 2.5. Sample Size

Based on the previous studies [20,32], we predicted the area under curve (AUC) of ROC curves for diagnosis to be 0.65 to 0.85. Subsequently, we calculated that with a sample of 30 patients per group, the study would have an 80% power to create an ROC curve with an AUC of 0.68 and a type I error of 5%. For the diagnosis of severity, we predicted the AUC of ROC curves for severity diagnosis to be 0.6 to 0.8 [15]. We calculated that with a sample of 98 patients, with 49 patients per group, the study would have 80% power to create an ROC curve with an AUC of 0.6 and a type I error of 5%. From these results, we collected 30 cases for the H group and a total of 100 cases for the M and S groups.

## 3. Results

### 3.1. Demographic and Clinical Characteristics

Table 2 summarizes the demographic and clinical data of each group. There was no significant difference in gender or age between the H, M, and S groups. As for comparisons between the two groups, the S group received more frequent injection therapy and was affected for a longer period than the M group. Fifty patients of the S group did not respond to conservative treatment and received surgical treatment; no patients of the M group required surgery.

### 3.2. MRI Scoring

In the combined evaluation, the median MRI score and 25–75 percentile were 1 (1–2), 3 (2–3), and 4 (3–4) in the H, M, and S groups, respectively. There was a significant difference among all groups: *p* < 0.001 for the H and M groups, *p* < 0.001 for the H and S groups, and *p* = 0.001 for the M and S groups (Figure 5). In the individual evaluation, the median MRI score and 25–75 percentile were 2 (1–2), 4 (3–5) and 6 (5–6) in the H, M, and S groups, respectively. There was a significant difference among all groups: *p* < 0.001 for the H and M groups, H and S groups, and for the M and S groups (Figure 6). Additionally, we described the distribution of the CET and LCL scores in each group in the Appendix A).

### 3.3. ROC Curve

In the ROC curve for diagnosis, shown in Figure 7, the AUC was 0.84 for combined evaluation (*p* < 0.001) and 0.86 for individual evaluation (*p* < 0.001). In the comparison of the evaluation methods, there was no significant difference in the AUC (*p* = 0.63). 

In the ROC curve for the diagnosis of clinical severity, shown in Figure 8, the AUC was 0.69 for the combined evaluation (*p* < 0.001) and 0.81 for the individual evaluation (*p* < 0.001). In the comparison of the evaluation methods, the AUC of the individual evaluation was significantly larger than that of the combined evaluation (*p* = 0.003).

### 3.4. Repeatability of MRI Scoring in This Study

The kappa values and their 95% confidence intervals for intra-observer agreement were 0.87 (0.85–0.90: *p* < 0.001) for examiner 1 and 0.86 (0.84–0.89: *p* < 0.001) for examiner 2. For inter-observer agreement between examiners 1 and 2, the kappa value was 0.84 (0.82–0.86: *p* < 0.001). The repeatability of MRI scoring was excellent.

## 4. Discussion

The most significant finding of this study was that individual MRI evaluations of the CET and the LCL improved the accuracy of the severity diagnosis of LE. Since MRI images reflect pathological change, we can accurately quantify pathological severity with detailed MRI scoring. Some of the studies investigating the relationship between clinical and MRI severity are commensurate with the results of this study. The literature with quantitative, individual evaluations of the CET and the LCL reported a positive association between clinical and MRI severity [7,15]; the studies without quantitative evaluation did not show this association [22]. Studies with quantitative, combined evaluations of the CET and the LCL reported conflicting conclusions [23,33]. This study suggests that the CET and the LCL should be individually evaluated using MRI to indicate the severity diagnosis.

Furthermore, we demonstrated the accuracy of MRI for the diagnosis of LE. According to the ROC curve for the diagnosis of LE, MRI had a high diagnostic capability, as reported in other tendinopathies [32]. Nevertheless, MRI is not always necessary for diagnosis since most patients with LE can be diagnosed based on physical findings. We should perform MRI only for patients who are refractory to conservative treatment. Differential diagnosis should be considered when MRI shows an absence or slight change in the signal on CET/LCL, e.g., the entrapment of the posterior antebrachial cutaneous nerve [34,35], radial tunnel syndrome, synovial fold disorder, posterolateral elbow instability, inflammatory disorders, cervical radiculopathy, and so on [36]. 

Meanwhile, further study is necessary to demonstrate the validity of MRI-positive findings for the severity diagnosis. Although MRI scores were higher in the group with higher clinical severity, there was some variation among cases. As Nirschl et al. showed, there is interindividual variation in symptoms and pathologic severity of LE [18,37]. Furthermore, some studies reported that psychological factors play a role in the intensity of the clinical symptoms in LE [38,39]. These findings indicate that the symptoms of LE are multifactorial, though based on pathological abnormalities. Thus, the cross-sectional study of the correlation between MRI scoring and clinical severity is necessarily limited. A longitudinal study should be conducted in the future to reveal the validity of positive MRI findings in LE. Since the literature suggests surgery in cases with severe pathology [18,37,40], MRI severity may predict the prognosis of conservative treatment. In particular, since posterolateral instability is reported to be associated with clinical severity [41,42], individual evaluation of LCL is significant. Overall, the findings of this study will be a basis for future research.

Our study had several strengths. Firstly, to our knowledge, this study used the highest-resolution MRI of any study to date; this allowed us to provide reliable data. The repeatability of MRI scoring was excellent. Secondly, we conducted a quantitative evaluation with a sufficient sample size, including the healthy group. Since previous reports have not quantitatively evaluated healthy subjects, we believe our data will serve as a basis for future MRI evaluations.

Although the study had many strengths, it also had some limitations. This study was retrospective, and we collected data on clinical symptoms from medical records. Although our treatment protocols and MRI indications are standardized at a single institution, there were some differences in the timing of MRI imaging in some cases. Additionally, our study included patients who received steroid injection therapy, which may have influenced the MRI findings or clinical assessment. Finally, we selected the subjects of the H group from a specific environment of medical coworkers in our hospital. Therefore, there is a possibility of selection bias that we could not predict. 

In conclusion, MRI individual evaluation of the CET and the LCL improved the accuracy of diagnosing the severity of LE. The CET and LCL should be evaluated individually to reflect the relationship of clinical severity to MRI severity accurately.

## Figures and Tables

**Figure 1 diagnostics-12-01871-f001:**
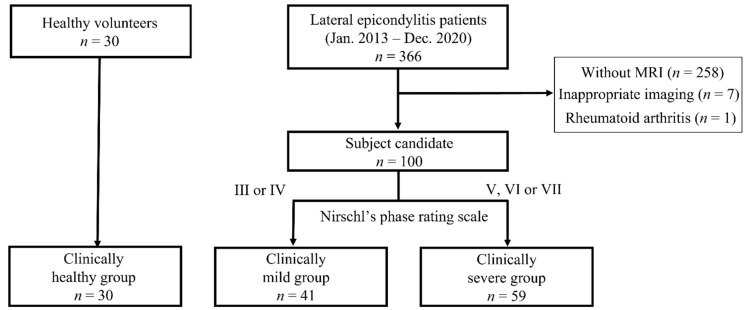
Flow diagram of patients included in the study.

**Figure 2 diagnostics-12-01871-f002:**
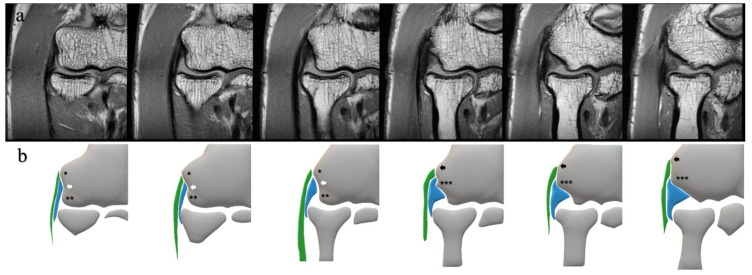
Consecutive MRI slices under PDWI of the unaffected lateral elbow (**a**) and corresponding schemas (**b**). These schemas represent bony landmarks and the CET and the LCL in this study. * superior tubercle; ** anterior tubercle; *** posterior tubercle; white arrow, intertubercular sulcus; black arrow, epicondylar ridge; green area, CET; blue area, LCL; MRI, magnetic resonance imaging; PDWI, proton-density-weighted imaging; CET, common extensor tendon; LCL, lateral collateral ligament.

**Figure 3 diagnostics-12-01871-f003:**
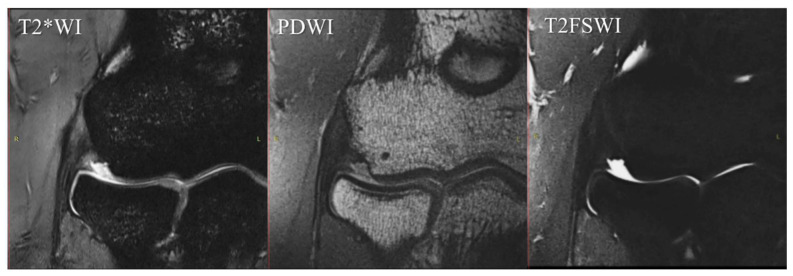
Differences in signal intensity of the CET/LCL complex by sequence. The unaffected elbow of a 32-year-old male. These images are same-level slices of the MRI coronal section in each sequence. In this case, T2*WI shows a high signal at the CET/LCL complex despite a complete low signal in the sequence of PDWI and T2FSWI. MRI, magnetic resonance imaging; T2*WI, T2*-weighted images; PDWI, proton-density-weighted images; T2FSWI, T2 fat-saturated weighted images; CET, common extensor tendon; LCL, lateral collateral ligament.

**Figure 4 diagnostics-12-01871-f004:**
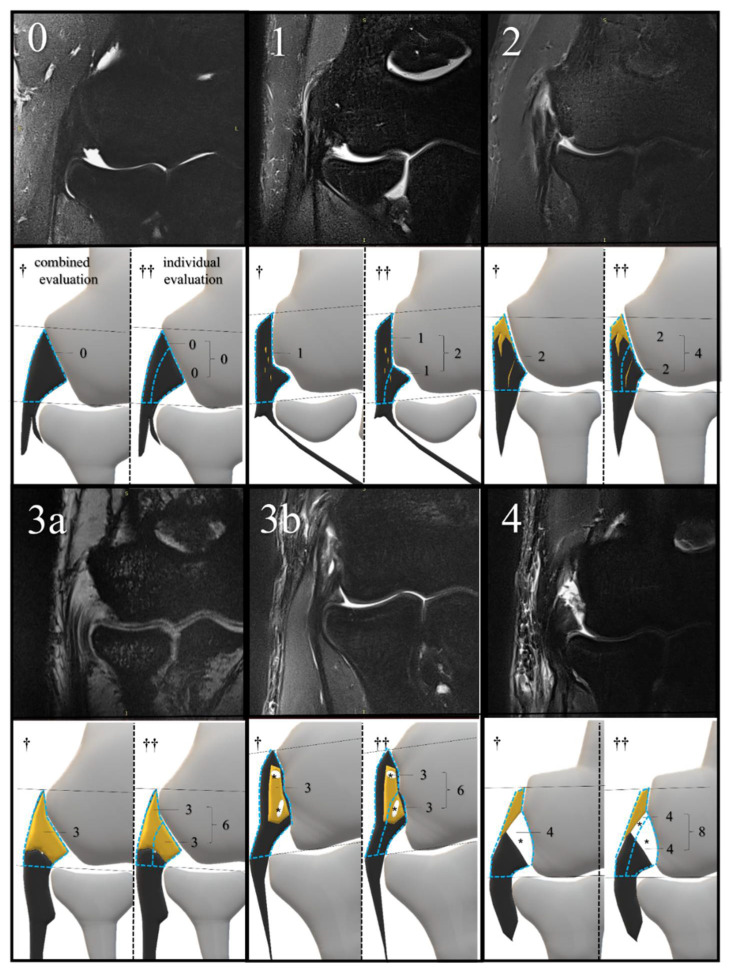
MRI examples of each score and its corresponding schema. The scoring criteria are as follows: (**0**) (normal), dark, linear low-signal structure without changes in signal intensity; (**1**) (mild degeneration), thickening or mild signal change below the signal intensity of the muscle; (**2**) (localized degeneration), high signal change above the signal intensity of the muscle, localized below 50% of the evaluation range; (**3a**) (extensive degeneration), high signal change above the signal intensity of the muscle, beyond 50% of the evaluation range; (**3b**) (partial tear), high signal change equivalent to joint fluid, within 75% of the tendon or ligament width; (**4**) (extensive tear), high signal change equivalent to joint fluid, more than 75% of the tendon or ligament’s width. The yellow area indicates degeneration; * tear; MRI, magnetic resonance imaging. The blue dotted line surrounds the region of interest for MRI scoring in each evaluation method. Black lines are auxiliary lines to determine the evaluation area, which runs parallel to the articular surface of the radial head. † combined evaluation; †† individual evaluation; yellow area, degeneration.

**Figure 5 diagnostics-12-01871-f005:**
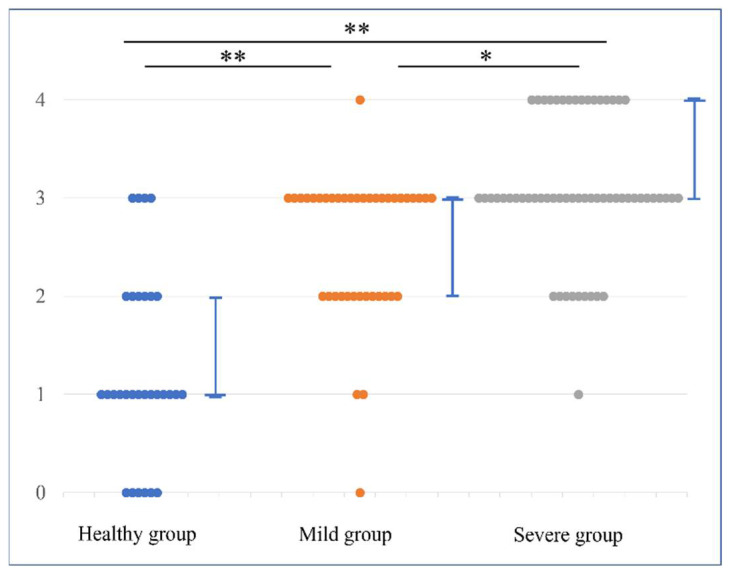
MRI scores in combined evaluation method. Dot plots show the differences in the distribution of MRI scores for each group. The MRI scores were valued as follows: healthy group < mild group < severe group, with significant differences. The line graph represents the median value and 25–75 percentile. * *p* < 0.05, ** *p* < 0.01.

**Figure 6 diagnostics-12-01871-f006:**
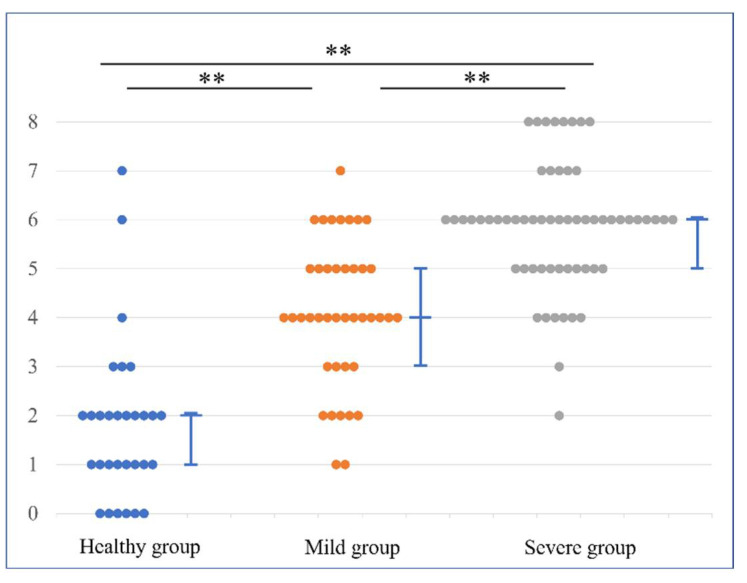
MRI scores in the individual evaluation method. Dot plots show the differences in the distribution of MRI scores for each group. The MRI scores were valued as follows: healthy group < mild group < severe group, with significant differences. The line graph represents the median value and 25–75 percentile. ** *p* < 0.01.

**Figure 7 diagnostics-12-01871-f007:**
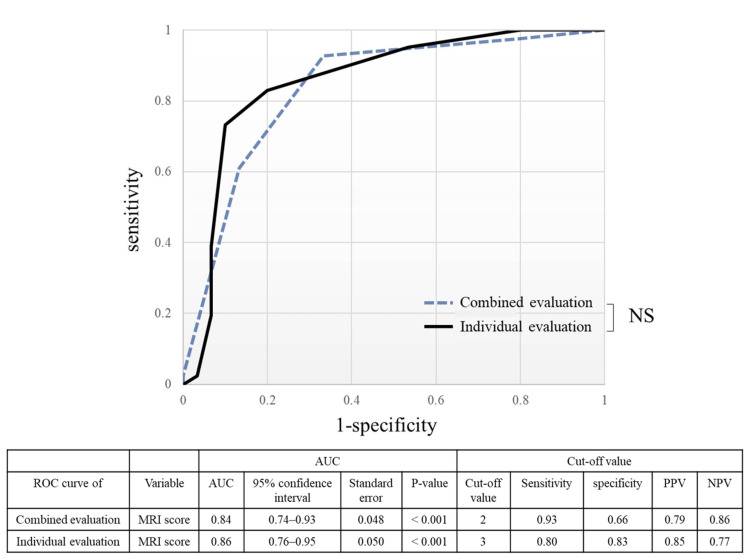
The ROC curve shows the capability of MRI to diagnose lateral epicondylitis for each evaluation method. In a comparison of AUC, there was no significant difference between the evaluation methods (*p* = 0.23). ROC curve, receiver operating characteristic curve; PPV, positive predictive value; NPV, negative predictive value; NS, not significant.

**Figure 8 diagnostics-12-01871-f008:**
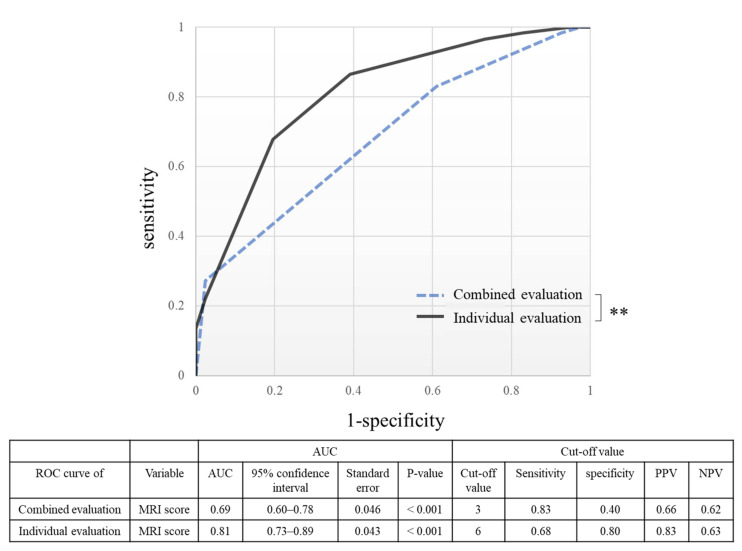
The ROC curve shows the capability of MRI to diagnose the clinical severity for each evaluation method. In a comparison of AUCs, individual evaluation was significantly superior to combined evaluation (*p* = 0.046). ** *p* < 0.05, ROC curve, receiver operating characteristic curve; PPV, positive predictive value; NPV, negative predictive value.

**Table 1 diagnostics-12-01871-t001:** Imaging parameters.

Sequence	T2*WI	PDWI	T2FSWI
Voxel size	0.2 × 0.2 × 1.5	0.2 × 0.2 × 1.5	0.2 × 0.2 × 1.5
Matrix	160 × 320	240 × 320	256 × 256
FOV	60 mm	60 mm	60 mm
Base resolution	320	320	256
Phase resolution	50%	50%	50%
Slice thickness	1.5 mm	1.5 mm	1.5 mm
TR	553.0 ms	553.0 ms	3000.0 ms
TE	24 ms	24 ms	94 ms
Bandwidth	180 Hz/Px	180 Hz/Px	145 Hz/Px
Flip angle	30	170	122

FOV: field-of-view, TR: repetition time, TE, echo time, T2*WI, T2*-weighted imaging; PDWI, proton-density-weighted imaging; T2FSWI, T2 fat-saturated weighted imaging.

**Table 2 diagnostics-12-01871-t002:** Demographic and clinical data of each group.

	Healthy Group	Mild Group	Severe Group	*p*-Value
Sex Male	12	21	31	*p* = 0.51
Female	18	20	28	
Age (y) †	49 (27–69)	49 (34–77)	49 (23–78)	*p* = 0.27
Injection therapy	-			*p* < 0.001
0		21	6	
1–2		17	29	
3≤		3	24	
unidentified		0	0	
Duration of pain (months) †	-	6.4 (2.1–81.0)	12.5 (1.4–133.1)	*p* = 0.032
0–1 month		0	0	
1–3 months		6	4	
3–6 months		12	12	
6–12 months		12	12	
>12 months		9	31	
Unidentified		2	0	
Required surgery ††	-	0/41	50/59	*p* < 0.001

† Data are presented as median (minimum–maximum); †† The surgical indication was for the patients with Nirschl’s clinical scale score of V or higher, who were resistant to the conservative treatment for at least 6 months.

## Data Availability

Not applicable.

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
