# Peer review of "Individual Evaluation of the Common Extensor Tendon and Lateral Collateral Ligament Improves the Severity Diagnostic Accuracy of Magnetic Resonance Imaging for Lateral Epicondylitis"

_diagnostics, 2022, doi:10.3390/diagnostics12081871_

Round 1
Reviewer 1 Report
1. Statistical analysis.
Line 160-162. Why did you use the Kolmogorov-Smirnov test? Is there a better Shapiro-Wilk test due to the small amount of data in the sample?
2. Table No. 2.
I do not understand why the p values are given for * and **. Subsequently in line 204: * p <0.05, ** p <0.01. Is that a bug. If not, please explain.
3. If the MRI scale is a point scale, the results should be presented as a median and min-max values.
4. What are the benefits for the patient from this type of diagnostics?
5. You conducted the study during January 2013 to December 2020, and the approvals for the study were in 2016 and 2021 (approved by the Institutional Review Board of the Mito Kyodo General Hospital (Study Number: 16-25, approved 7 September 2016) and Takahagi Kyodo General Hospital (Study Number: 10, approved 3 March 2021)). Please explain.
Author Response
Thank you very much for your review and valuable suggestions.
1. Statistical analysis.
Line 160-162. Why did you use the Kolmogorov-Smirnov test? Is there a better Shapiro-Wilk test due to the small amount of data in the sample?
Thank you for pointing this out. We have confirmed that the Shapiro-Wilk test does not follow a normal distribution. We have changed this description.
- Table No. 2.
I do not understand why the p values are given for * and **. Subsequently in line 204: * p <0.05, ** p <0.01. Is that a bug. If not, please explain.
Thank you for pointing this out. This description is unnecessary as the specific values are listed; therefore, we have removed it.
- If the MRI scale is a point scale, the results should be presented as a median and min-max values.
Thank you for pointing this out. Considering the characteristics of the data, we have changed the data presentation to the median (25-75 percentile).
- What are the benefits for the patient from this type of diagnostics?
This is indeed the question we are trying to answer in this study of MRI for lateral epicondylitis. As we described in the discussion, positive MRI findings for lateral epicondylitis are currently of limited clinical value. Further longitudinal studies investigating positive MRI findings and prognosis of conservative treatment are needed to determine the full clinical value of MRI. This study presents the basic findings of quantitative MRI evaluation methods which will serve as a basis for future research.
- You conducted the study during January 2013 to December 2020, and the approvals for the study were in 2016 and 2021 (approved by the Institutional Review Board of the Mito Kyodo General Hospital (Study Number: 16-25, approved 7 September 2016) and Takahagi Kyodo General Hospital (Study Number: 10, approved 3 March 2021)). Please explain.
We obtained IRB approval for an observational study on LE patients (from 2013 to 2020) at Mito Kyodo Hospital in 2016. In addition, we obtained IRB approval for MRI imaging on healthy subjects at Takahagi Kyodo Hospital in 2021. These are affiliated hospitals and have the same MRI equipment, micro coils, and imaging conditions.
Reviewer 2 Report
This is a well-conducted study. The study design is robust. Besides the MRI pictures, the authors also added meticulously drawn illustration to improve the readership. I would like to congratulate the great efforts from the authors to accomplish this work.
I have some minor suggestions:
First, the prevalence of lateral epicondylitis should be given in the introduction.
Second, ultrasound is commonly used for the examination of lateral epicondylitis. The authors are suggested to add a paragraph to compare both modalities.
Third, please provide the number of the IRB approval in Section 2.1.
Fourth, the entrapment of posterior antebrachial cutaneous nerve could mimic lateral epicondylitis. The authors can mention it. The following two publications can be mentioned:
https://pubmed.ncbi.nlm.nih.gov/30469370/
https://link.springer.com/article/10.1007/s00256-020-03594-7
Author Response
Thank you very much for your review and valuable suggestions.
First, the prevalence of lateral epicondylitis should be given in the introduction.
Thank you for pointing this out. I have added a description of the prevalence of lateral epicondylitis.
Second, ultrasound is commonly used for the examination of lateral epicondylitis. The authors are suggested to add a paragraph to compare both modalities.
Thank you for pointing this out. I have added text to state that the MRI can be contrasted with the ultrasound.
Third, please provide the number of the IRB approval in Section 2.1.
Thank you for pointing this out. we have added the IRB number to Section 2.1.
Fourth, the entrapment of posterior antebrachial cutaneous nerve could mimic lateral epicondylitis. The authors can mention it. The following two publications can be mentioned:
https://pubmed.ncbi.nlm.nih.gov/30469370/
https://link.springer.com/article/10.1007/s00256-020-03594-7
Thank you for your valuable suggestions. This was indeed an interesting and informative study; I have added it to the discussion as a differential disease.